# Downregulation of the *GhROD1* Gene Improves Cotton Fiber Fineness by Decreasing Acyl Pool Saturation, Stimulating Small Heat Shock Proteins (sHSPs), and Reducing H_2_O_2_ Production

**DOI:** 10.3390/ijms252011242

**Published:** 2024-10-19

**Authors:** Bo Ding, Bi Liu, Xi Zhu, Huiming Zhang, Rongyu Hu, Silu Li, Liuqin Zhang, Linzhu Jiang, Yang Yang, Mi Zhang, Juan Zhao, Yan Pei, Lei Hou

**Affiliations:** 1College of Agronomy and Biotechnology, Southwest University, Chongqing 400715, China; 15086655435@163.com (B.D.); morningrain7@sohu.com (B.L.); zhuxi0603@email.swu.edu.cn (X.Z.); zhang15828612304@163.com (H.Z.); 18509175153@163.com (R.H.); lisilu20010817@outlook.com (S.L.); zhangliuqin0715@163.com (L.Z.); linzhuhh@163.com (L.J.); neo5334@163.com (Y.Y.); selenazm@swu.edu.cn (M.Z.); zhaojuan0920@swu.edu.cn (J.Z.); peiyan3@swu.edu.cn (Y.P.); 2Chongqing Key Laboratory of Application and Safety Control of Genetically Modified Crops, Southwest University, Chongqing 400715, China

**Keywords:** cotton fiber development, phosphatidylcholine diacylglycerol cholinephosphotransferase (PDCT), *GhROD1*, linoleic acid, H_2_O_2_, small heat shock proteins, cellulose synthesis, *Gossypium hirsutum* L.

## Abstract

Cotton fiber is one of the most important natural fiber sources in the world, and lipid metabolism plays a critical role in its development. However, the specific role of lipid molecules in fiber development and the impact of fatty acid alterations on fiber quality remain largely unknown. In this study, we demonstrate that the downregulation of *GhROD1*, a gene encoding phosphatidylcholine diacylglycerol cholinephosphotransferase (PDCT), results in an improvement of fiber fineness. We found that *GhROD1* downregulation significantly increases the proportion of linoleic acid (18:2) in cotton fibers, which subsequently upregulates genes encoding small heat shock proteins (sHSPs). This, in turn, reduces H_2_O_2_ production, thus delaying secondary wall deposition and leading to finer fibers. Our findings reveal how alterations in linoleic acid influence cellulose synthesis and suggest a potential strategy to improve cotton fiber quality by regulating lipid metabolism pathways.

## 1. Introduction

Cotton fiber, the primary natural fiber source for the global textile industry, constitutes around 35% of the world fiber production [1]. The economic significance of the cotton textile industry is substantial, with a global market value exceeding USD 600 billion annually [2]. Cotton fibers develop as single cells, exhibiting remarkable uniformity and synchronicity during their growth. When fully matured, the fibers achieve a length-to-diameter ratio ranging from 1000 to 3000. Uniquely, mature cotton fibers consist of more than 95% cellulose by dry weight, with very low lignin content, distinguishing them from most other plant cell walls. These characteristics make cotton fibers an ideal model for investigating plant cell differentiation, elongation, and secondary cell wall development [3]. Fiber development can be broadly categorized into five overlapping stages: initiation, elongation, transition, secondary cell wall (SCW) biogenesis, and dehydration maturation. Among them, the elongation stage determines final fiber length, while the SCW biogenesis stage determines fiber fineness and strength. The transition stage, which marks the shift from primary to secondary cell wall synthesis, significantly influences both fiber length and cell wall thickness [4].

Lipids are vital for organisms, serving as key components of biological membranes that create a hydrophobic barrier to preserve the intracellular environment necessary for metabolic processes. They also function as energy storage compounds essential for cell growth and development as well as signaling molecules that regulate various cellular processes [5]. Accumulating investigations show that during fiber initiation and elongation, the lipid metabolism pathway is significantly enhanced [6,7,8,9]. However, the precise mechanisms of lipid metabolism in cotton fiber development remain poorly understood, and the specific roles of various lipid species in these processes await to be elucidated.

The rapid synthesis of cytoplasmic and vacuolar membranes during fiber elongation increases the cellular demand for glycerophospholipids [10,11]. Glycerophospholipids, essential components of cell membranes, are classified into six major types: phosphatidylcholine (PC), phosphatidylethanolamine (PE), phosphatidylinositol (PI), phosphatidylserine (PS), phosphatidic acid (PA), and phosphatidylglycerol (PG) [12,13].

PC is a major component of cell membrane phospholipids, serving as a substrate for fatty acid desaturation [14]. In cotton fibers, PC is the most abundant lipid class during the elongation and secondary wall synthesis stages [10]. However, the specific function of PC in cotton fiber development remains largely unknown. Phosphatidylcholine cholinephosphotransferase (PDCT) catalyzes the transfer of the phosphocholine headgroup between PC and diacylglycerol (DAG), facilitating the exchange of fatty acyl groups between these lipid molecules [15]. In plants, the impact of PDCT, also known as ROD1, on fatty acid composition in seeds has been well-documented [16,17,18,19]. However, the influence of ROD1-mediated fatty acid alterations on cotton fiber development and fiber quality remains unexplored.

To investigate the function of PDCT in lipid metabolism in fiber cells and assess its role in fiber quality and yield traits, we cloned an *AtROD1* homologous, *GhROD1*, from upland cotton (*Gossypium hirsutum*). We demonstrated that downregulation of *GhROD1* resulted in an increase in C18:2 levels in fibers. Importantly, consecutive three-year field trails showed that the fineness of the *GhROD1*-downregulated fibers improved significantly. We further revealed that the elevated C18:2 level promoted the transcription of genes encoding small heat shock proteins (sHSPs), which in turn decreased the production of H_2_O_2_. This reduction in H_2_O_2_ inhibited secondary cell wall synthesis, ultimately resulting in finer fibers. A working model for the linoleic acid (C18:2)-induced inhibition of second cell wall deposition is proposed.

## 2. Results

### 2.1. Downregulation of GhROD1 Significantly Increases Linoleic Acid in Fibers

To investigate the role of fatty acids (FAs) in fiber development, we identified and cloned *GhROD1*, which encodes phosphatidylcholine diacylglycerol choline phosphotransferase (PDCT), from upland cotton. qRT-PCR results indicated that *GhROD1* transcripts were abundant in cotton petals, boll shells, ovules, and fibers (Figure 1A). During fiber development, *GhROD1* transcript levels steadily increased from 6 to 20 days post-anthesis (DPA), followed by a sharp decline after 20 DPA (Figure 1B).

To elucidate the role of *GhROD1* in fiber development, we generated its overexpressing and RNA interference (RNAi) lines in upland cotton via *Agrobacterium*-mediated transformation (Appendix A). The fatty acid composition GC-FID assay showed that in the fibers at 20 DPA of *GhROD1*-RNAi lines Ri3 and Ri4, palmitic acid (C16:0) and linolenic acid (C18:3) levels were slightly lower than in the wild-type, while stearic acid (C18:0) and oleic acid (C18:1) levels were comparable. However, linoleic acid (C18:2) content was significantly higher in the transgenic lines Ri3 and Ri4 (17.21 ± 0.12% and 15.25 ± 0.12%, respectively) compared to the wild-type (12.91 ± 0.19%) (Figure 2A). In the fibers of overexpressing lines OE17 and OE25, C18:2 was slightly lower and C16:0 was slightly higher; while C18:0 (2.37 ± 0.16% and 2.67 ± 0.23%) was significantly higher and C18:1 (2.90 ± 0.02% and 2.71 ± 0.05%) was significantly lower compared to the wild-type (C18:0 = 1.33 ± 0.02% and C18:1 = 4.26 ± 0.16%, respectively).

To investigate whether alterations in fatty acid composition extended to the cell membrane, glycerophospholipid (GPL) species were extracted from fibers at 20 DPA and quantified. Analysis revealed significant changes in phosphatidylcholine (PC) and phosphatidylethanolamine (PE) species. PC34:3, the most abundant PC species, was significantly reduced in Ri3 and Ri4 fibers, while PC34:2 was increased significantly only in Ri4 fibers. Among PE species, the most abundant, PE34:3, was significantly reduced in Ri4 fibers. PE34:2, a less abundant PE species, displayed similar changes to PC34:2 in Ri4 fibers (Figure 2C). However, diacylglycerol (DAG) species exhibited more complex alterations. DAG34:2 (16:0–18:2) increased in the Ri3 and Ri4 fibers, while other components remained largely unchanged. In the OE17 and OE25 fibers, DAG34:3 (16:0–18:3) increased, while DAG36:2 (18:0–18:2), DAG36:3 (18:1–18:2), and DAG36:3 (18:0–18:3) decreased with no significant changes in other species (Figure 2D). Other GPLs, including phosphatidylglycerol (PG), phosphatidic acid (PA), and phosphatidylserine (PS), showed no significant differences between the transgenic and wild-type fibers (Appendix A).

### 2.2. Downregulating GhROD1 Improves Fiber Fineness

To assess the impact of altered fatty acids resulted from *GhROD1* regulation on fiber quality, we tested fiber qualities of the transgenic cotton lines across three consecutive generations. Notably, the *GhROD1*-downregulated fibers exhibited significant improvements in fiber fineness, as indicated by reduced micronaire (MIC) values. Specifically, the Ri3 fibers displayed a 12% to 17% reduction in MIC values (Table 1). Additionally, the *GhROD1*-downregulated fibers, particularly the Ri3 fibers, showed increased fiber length and strength. However, no significant change in fiber quality was observed in *GhROD1*-upregulated fibers (Table 1).

The MIC value is a key indicator of both fiber fineness (linear density) and maturity (the degree of secondary cell wall development) [20]. To further support the fineness improvement in *GhROD1*-downregulated fibers, we examined cross-sections of mature fibers (Figure 3). The thicknesses of cell walls of fibers in the downregulated Ri3 and Ri4 lines were 3.71 ± 0.42 and 4.29 ± 0.30 μm, respectively, obviously thinner than that of the wild-type (4.86 ± 0.38 μm, Figure 3F), confirming that the *GhROD1*-downregulating fibers became thinner.

### 2.3. Downregulation of GhROD1 Leads to Upregulation of sHSPs and Downregulation of Secondary Wall Synthesis-Associated Genes

To investigate the role of *GhROD1* in secondary wall synthesis, we performed a transcriptome analysis of fibers at 16 DPA from both the Ri3 line and the wild-type. A total of 81,931 transcripts were obtained and 70,478 hypothetical genes were annotated. A total of 540 differentially expressed genes (DEGs) were identified in the Ri3 fibers. Among them, 286 were upregulated and 254 were downregulated (Figure 4A).

To accurately determine the link between these DEGs and fiber fineness, we performed Gene Ontology (GO) term enrichment analysis. The analysis revealed that the DEGs were primarily associated with primary and secondary cell wall biogenesis and biosynthesis, metabolism of cell wall components, osmotic and heat stress response genes, and unfolded protein binding (Figure 4B). To validate the transcriptomic results, we further analyzed GO-enriched DEGs related to plant cell wall synthesis (Appendix A). We selected *GhCesA4B*, *GhCesA7B*, and *GhCesA8B*, genes involved in fiber secondary wall synthesis, for qRT-PCR analysis. In the fibers at 16 DPA of the Ri3 and Ri4 lines, the expression levels of these genes were significantly reduced, while no significant changes were observed in the OE17 and OE25 lines compared to the wild-type (Figure 5B).

We also mapped the DEGs to the Kyoto Encyclopedia of Genes and Genomes (KEGG) database. The DEGs were significantly enriched in two KEGG pathways: protein processing in the endoplasmic reticulum and phenylpropanoid biosynthesis (Figure 4C). The majority of the endoplasmic reticulum protein processing genes encode small heat shock proteins (sHSPs), such as *GhHSP17.6B/GhHSP17.7C*, *GhHSP17.8A/B*, *GhHSP17.9A/C/F*, *GhHSP18.1B/C*, *GhHSP18.2B*, *GhHSP18.3B/D*, *GhHSP18.4*, and *GhHSP20.7A* (Appendix A). qRT-PCR results confirmed that the expression of these genes was significantly upregulated in the Ri3 and Ri4 fibers, while they remained unchanged in the OE17 and OE25 lines (Figure 5A). These data suggest a connection between *GhROD1* downregulation, sHSPs-related protein processing, and cell wall biosynthesis.

To support this connection, we cultured wild-type ovules on the medium containing linoleic acid. The presence of linoleic acid resulted in a significant upregulation of several *sHSP* genes, including *GhHSP17.6B/GhHSP17.7C*, *GhHSP17.9A/C/F*, *GhHSP18.1B/C*, *GhHSP18.3B/D*, *GhHSP18.4*, and *GhHSP20.7A*, while there was a significant downregulation of the cellulose synthase genes *GhCesA4B*, *GhCesA7B*, and *GhCesA8B* (Figure 5D). Interestingly, adding dithiothreitol (DTT), an ER stress inducer, to the medium also resulted in the upregulation of *sHSPs* (Figure 5C).

### 2.4. Small HSPs Inhibit Secondary Wall Synthesis by Reducing ROS Levels in Fibers

Previous studies demonstrated that upregulation of *sHSPs* can reduce the accumulation of reactive oxygen species (ROS) [21,22,23]. This prompted us to investigate whether the overexpression of cotton *sHSPs* could similarly inhibit ROS production. Upon transiently expressing *GhHSP17.6B*, *GhHSP17.9F*, and *GhHSP18.4* in tobacco, we observed a significant reduction in H_2_O_2_ levels in the treated leaves (Figure 6A). Moreover, in Ri3 and Ri4 fibers, where the *sHSPs* genes were upregulated, the H_2_O_2_ content was significantly lower than that of wild-type (Figure 6B). Examining the H_2_O_2_ content in the *Arabidopsis rod1* siliques, we also observed a similar decrease in the mutant (Figure 6C). Importantly, culturing the wild-type ovules in the media containing linoleic acid (5 μM and 50 μM) resulted in a decrease in H_2_O_2_ content in the treated fibers (Figure 6D).

Previous research established that H_2_O_2_ acts as a signaling molecule that promotes cellulose deposition in plant cell wall [24,25,26]. When we treated wild-type ovules with diphenyleneiodonium (DPI) to block superoxide production, a significant reduction in H_2_O_2_ content was observed (Figure 6E). Concurrently, the expression of cellulose synthase genes involved in secondary wall deposition, *GhCesA4B*, *GhCesA7B*, and *GhCesA8B*, was significantly downregulated (Figure 6F). Taken together, these results indicate that downregulation of *GhROD1* can upregulate *sHSPs*, which in turn inhibits the secondary wall synthesis by reducing ROS levels in fibers.

## 3. Discussion

### 3.1. GhROD1-Mediated DAG-to-PC Interconversion May Contribute to Fatty Acid Desaturation in Cotton Fibers

Following initiation, cotton fibers undergo rapid elongation to ultimately form lint. The period between 16 and 20 days post-anthesis (DPA) represents a critical transition from primary to secondary cell wall synthesis. Analysis of the fatty acid composition of fibers during this period revealed linolenic acid (18:3) and palmitic acid (16:0) as the two most abundant fatty acids (Figure 2A), consistent with previous findings [10,11] and highlighting the potential importance of 18:3 in fiber development. In fiber cells, as in most plant cells, 16:0 and 18:1 are the major fatty acids synthesized in plastids and exported to the endoplasmic reticulum (ER). These fatty acids undergo a series of enzymatic reactions to form phosphatidic acid (PA), which is then dephosphorylated to yield DAGs. DAGs are subsequently used to synthesize PC and PE. Finally, the 18:1 acyl chain is modified by FAD2 and FAD3 to produce 18:2 and 18:3, respectively [10,27]. Notably, DAGs can also be converted to PCs by PDCT, and the acyl chains of PCs are further desaturated by FADs in the ER to reduce overall fatty acid saturation and meet cellular demands [15,28]. Our study revealed high expression of *GhROD1*, encoding PDCT, in developing fibers from 14 to 20 DPA, in addition to its expression in other tissues. Furthermore, the encoded PDCT protein is primarily localized in the ER (Appendix A). These findings suggest that *GhROD1* may facilitate the interconversion of DAG and PC to increase the unsaturation of the acyl pool during fiber development, potentially representing a complementary pathway to fatty acid desaturation. Consistent with this hypothesis, downregulation of *GhROD1* led to an increase in PC 34:2 (16:0–18:2) and PE 34:2 (16:0–18:2) content, while PC 34:3 (16:0–18:3) and PE 34:3 (16:0–18:3) content decreased. Moreover, the 18:3 content in total fatty acids was reduced (Figure 2A). Conversely, upregulation of *GhROD1* increased PC 36:6 (18:3–18:3) content in OE17 and OE25 fibers (Figure 2B), indicating that PDCT plays a key role in increasing polyunsaturated fatty acids (PUFAs) in fibers, similar to its function in *Arabidopsis* [16]. However, while *GhROD1* overexpression increased 18:3 content in fatty acids by 2.4–8.4% and downregulation reduced it by 2.8–6.5% (Figure 2A), the magnitude of these changes was relatively modest. This suggests that while increased fatty acid unsaturation via the DAG-to-PC interconversion pathway mediated by PDCT may occur in fibers, it likely represents a complementary rather than primary route to achieving optimal fatty acid composition during fiber development.

Given the observed alterations in PC and PE fatty acid saturation in the fibers of Ri3 and Ri4 lines, a corresponding change in DAG saturation might be expected. However, our results revealed a less pronounced pattern of DAG variation (Figure 2D). DAG plays a central role in plant lipid metabolism, and its synthesis involves multiple reactions occurring in distinct subcellular compartments, including the ER, lipid droplets (LDs), and plasma membrane [29]. DAG serves as a precursor for triacylglycerol (TAG) synthesis, the important role of which is to storage energy, and serves as an intermediate in the synthesis of glycerophospholipids, such as PC and PE to participate in various lipid metabolic pathways [27,30,31]. Consequently, alterations in DAG saturation are likely influenced by other lipid metabolic pathways, providing a possible explanation for the complex changes in DAG saturation observed in our study.

### 3.2. Alterations in Phospholipid Fatty Acid of Fibers May Activate the ER Stress Response

The endoplasmic reticulum (ER) plays a crucial role in lipid and protein biosynthesis, serving as the primary site for phosphatidylcholine (PC) and phosphatidylethanolamine (PE) synthesis [32,33]. PC is the most abundant phospholipid in the ER membrane of eukaryotes [34,35]. Inhibition of PC biosynthesis is known to induce ER stress [36,37,38]. In this study, downregulation of *GhROD1* not only increased fatty acid saturation but also altered PC and PE species in cotton fibers. Do these changes in PC and PE species induce ER stress in fiber cells? Our KEGG pathway enrichment analysis provides a potential clue for answering this question. The analysis identified protein processing in the ER as a significantly affected pathway, which is involved in the upregulation of genes from the small heat shock protein (sHSP) family (Figure 4), suggesting the activation of a stress response. In vitro supplementation with linoleic acid (18:2) resulted in a similar upregulation of several *sHSP* genes in fibers (Figure 5), as did the treatment with the ER stress inducer DTT. These results support a link between downregulation of *GhROD1*, encoding PDCT, and ER stress.

### 3.3. The Upregulation of sHSPs Can Reduce the Accumulation of ROS in Cells and Inhibit the Synthesis of Secondary Wall

Now the question is how does the downregulating PDCT-induced ER stress affect secondary cell wall biosynthesis in fibers? Heat shock proteins (HSPs) are a diverse group of proteins induced by various cellular stresses and are categorized into distinct families based on homology and molecular weight, including HSP100, HSP90, HSP70, HSP60, and small heat shock proteins (sHSPs) [39]. sHSPs, with molecular masses ranging from 12 to 42 kDa, are known for their protective roles in a variety of crops [40]. Previous studies showed that upregulation of some *sHSPs* could reduce H_2_O_2_ production. Overexpression of the maize HSP, *ZmHSP16.9*, has been shown to decrease H_2_O_2_ levels in tobacco under heat stress [41]. Similarly, overexpression of tomato *SlHSP17.7* mitigated H_2_O_2_ accumulation under cold stress [42], and *Arabidopsis* plants overexpressing *ClHSP20* from *Camellia limonia* exhibited reduced H_2_O_2_ levels [23]. Likely, our research demonstrated that transient expression of *GhHSP17.6B*, *GhHSP17.9F*, and *GhHSP18.4* in tobacco leaves significantly reduced H_2_O_2_ accumulation (Figure 6A).

H_2_O_2_ is recognized as a signaling molecule that initiates secondary cell wall (SCW) formation in cotton fibers [43]. Transcriptome and metabolome analyses of upland cotton suggest that fiber cell wall thinning may be associated with enhanced antioxidant capacity, emerging as a strategy for improvement of fiber finesses by reducing reactive oxygen species (ROS) levels to delay SCW synthesis [44]. Mutations in *GhMDHAR1AT/DT* and *GhMDHAR2AT/DT* resulted in a reduction in H_2_O_2_ content, thus downregulating genes related to secondary wall synthesis and delaying the synthesis process [26]. In this study, downregulation of *GhROD1* in cotton led to a marked decrease in H_2_O_2_ content (Figure 6B) and significant improvement in fiber finesses. Interestingly, the fiber strength of *GhROD1*-downregulated line Ri3 was simultaneously increased. Our findings reveal how the mechanism of downregulation of *GhROD1* affects secondary cell wall synthesis through enhancing *sHSP* expression and H_2_O_2_ production. The study also suggests a strategy for improving cotton fiber quality by the regulation of the lipid metabolism pathway.

Our findings uncover the mechanism by which downregulation of *GhROD1* affects secondary cell wall synthesis by inducing ER stress, which in turn enhances *sHSP* expression and H_2_O_2_ production (Figure 7). This study also suggests a potential strategy for improving cotton fiber quality through the regulation of lipid metabolism pathways.

## 4. Materials and Methods

### 4.1. Construction of Vectors

Construction of the pCB2012-*GhROD1* overexpression vector: The coding sequence of *GhROD1* (Gene ID: Gh_A06G1621/Gh_D06G1990) was amplified from upland cotton fiber cDNA. The purified PCR product was ligated into the pEASY-Blunt Cloning Vector (Transgen, China). The cloning vector containing *GhROD1* was digested with *Bam*HI and *Xba*I to release the gene fragment. The pCB2012 vector was digested with *Bam*HI and *Spe*I to generate a compatible vector backbone. The *GhROD1* fragment was ligated into the linearized pCB2012 vector using the Rapid DNA Ligation Kit (Roche, Basel, Switzerland), creating the 35S-*GhROD1* overexpression construct.

Construction of the pCB2012-*GhROD1*-RNAi vector: Sense, intron, and antisense fragments of *GhROD1* were sequentially fused via PCR to create a hairpin structure-forming sequence. The fusion fragment was cloned into the pUCm-T vector (Sangon, Shanghai, China). Both the pUCm-T containing the *GhROD1*-RNAi fragment and pCB2012 vectors were digested with *Kpn*I and *Bam*HI. The *GhROD1*-RNAi fragment was ligated into the linearized pCB2012 vector using the Rapid DNA Ligation Kit (Roche, Switzerland), resulting in the 35S-*GhROD1*-RNAi construct.

The coding sequences of *GhHSP17.6B* (Gene ID: Gh_D08G0057), *GhHSP17.9F* (Gene ID: Gh_D12G1364), and *GhHSP18.4* (Gh_D03G1268) were amplified from wild-type fiber cDNA. Each amplified fragment was individually cloned into the pCB2016 vector, which was linearized with *Eco*RI and *Spe*I, using the ClonExpress II One Step Cloning Kit (Vazyme, Nanjing, China). This process generated three separate 35S-*GhHSP* overexpression constructs: 35S-*GhHSP17.6B*, 35S-*GhHSP17.9F*, and 35S-*GhHSP18.4*.

### 4.2. Plant Materials and Growth Conditions

Upland cotton (*Gossypium hirsutum* L. cv. Jimian14) was used for transformation. The genetic transformation of cotton followed the protocol described by Luo et al. [45]. *GhROD1* transgenic cotton plants were grown in experimental plots at Southwest University.

### 4.3. RNA Extraction and qRT-PCR

Total RNA was isolated from plant samples using the RN09-EASY Spin Plant RNA Rapid Kit (Aidlab Biotech, Beijing, China). Subsequently, complementary DNA (cDNA) synthesis was carried out using the PrimeScript RT Reagent Kit with gDNA Eraser (TaKaRa, Beijing, China). qRT-PCR was performed in accordance with the methodology outlined by Yang et al. [46]. *GhHis3* (Accession number: AF024716) served as reference. The specific primers employed are detailed in Appendix A.

### 4.4. Fatty Acid Analysis

Cotton fiber lipid extraction: Lipids were extracted from cotton fibers using a modified method based on Liu et al. [11]. All solvents used contained 0.01% butylated hydroxytoluene (BHT) to prevent lipid oxidation. Isopropanol was preheated to inactivate phospholipase D. Briefly, 100 mg of fresh cotton fiber was incubated with 3 mL of preheated isopropanol at 75 °C for 15 min. Heptadecanoic acid (C17:0) was added as an internal standard. Subsequently, 0.6 mL of water and 1.5 mL of chloroform were added, and the mixture was incubated at 4 °C with shaking at 200 rpm for 2 h. The lipid extracts were transferred to a new polytetrafluoroethylene tube. An additional 4 mL of a methanol/chloroform (1:2, *v*/*v*) mixture was added, and this extraction step was repeated twice. The combined lipid extracts were mixed with 1 mL of 1 M KCl, vortexed thoroughly, and centrifuged at 2000 rpm for 10 min. The resulting precipitate was transferred to a separate glass tube, dried, and stored at −80 °C.

Cotton fiber fatty acid (FA) extraction: Cotton fiber FAs were extracted with 2.5% H_2_SO_4_ in methanol (*v*/*v*), following the methodology outlined by Liu et al. [11].

Gas chromatography with flame ionization detection (GC-FID) analysis: Fatty acid methyl esters (FAMEs) in hexanes were analyzed by GC-FID using the following temperature program. Initial temperature: 100 °C for 1 min. Ramp 1: 20 °C/min to 170 °C, hold for 1 min. Ramp 2: 5 °C/min to 220 °C, hold for 12 min.

### 4.5. Lipidomic Analyses

The lipid was extracted by the same methods without internal standard. Lipidomic analyses were conducted by the National Key Laboratory of Crop Genetic Improvement at Huazhong Agricultural University, using direct infusion via LC-ESI-MS/MS, as described by Han and Gross [47].

### 4.6. Fiber Cell Wall Thickness and Quality Assessment

Cell wall thickness of mature fibers was evaluated using the method described by Han et al. [48]. Mature seed cotton was harvested from both transgenic and wild-type plants on the same day. Samples were dried and ginned before being submitted to the Supervision Inspection and Test Center of Cotton Quality, Ministry of Agriculture and Rural Affairs of China, for analysis. Fiber quality parameters were measured using a high-capacity integrated cotton fiber tester (model HVI-1000). The following properties were assessed: fiber length, uniformity index, fiber strength, micronaire value, and elongation ratio. Three independent samples were prepared for each material (transgenic and wild-type). Statistical analysis of the fiber quality data was performed using Student’s *t*-test to compare differences between transgenic and wild-type plants.

### 4.7. RNA Sequencing and Transcriptome Analysis

Total RNA was extracted from fibers 16 days post-anthesis (DPA) of both *GhROD1*-RNAi transgenic lines and wild-type plants using the MJZol total RNA extraction kit (TRIzol). Three biological replicates were included for each sample. RNA-seq libraries were constructed using the Illumina^®^ Stranded mRNA Prep Kit (Illumina, San Diego, CA, USA) with 1 μg of total RNA per sample. Paired-end sequencing (150 bp reads) was performed on the Illumina Novaseq 6000 platform by Shanghai Majorbio Bio-pharm Technology Co., Ltd. (Shanghai, China). Raw sequencing reads were deposited in the NCBI Sequence Read Archive (SRA) database under Accession Number SRP525959. Reads were aligned to the *Gossypium hirsutum* (AD1) TM-1 genome NAU-NBI_v1.1 as the reference genome. Data analysis was performed using Majorbio Cloud Platform (www.majorbio.com). Differentially expressed genes (DEGs) between *GhROD1*-RNAi lines and wild-type were identified based on an adjusted *p*-value threshold of <0.05 and a fold change criterion of ≥ 2.

### 4.8. Ovule Culture and Treatment

Cotton ovules at anthesis were harvested and incubated in BT medium at 32 °C for 12 days, as detailed in previous work [49]. Subsequently, the ovules were transferred to BT medium supplemented with various concentrations of linoleic acid (C18:2) or 1 μM diphenyleneiodonium (DPI) and further cultured for an additional 4 days. The BT medium was adjusted with the amount of DMSO (dimethyl sulfoxide) and 0.01% NP-40 equivalent to that used to dissolve linoleic acid (C18:2), or DMSO equivalent to that used to dissolve DPI as indicated, which was used as the mock control, respectively.

Dithiothreitol (DTT) treatment: Cotton ovules were collected at anthesis and cultured in BT medium at 32 °C for 4 h in the presence of 100 μM DTT. BT medium containing an equivalent amount of DMSO as used to dissolve DTT served as the mock control.

### 4.9. Statistical Data Analysis

The software used for statistical data analysis was GraphPad Prism 8.0 and SPSS 16.0. The heat map was built by Tbtools-II v2.119 [50].

## 5. Conclusions

This study identified *GhROD1*, a gene encoding phosphatidylcholine: diacylglycerol cholinephosphotransferase, in cotton. Downregulation of *GhROD1* increased linoleic acid (C18:2) content in fiber cells, which was accompanied by an increase in C18:2-containing phosphatidylcholine (PC) and phosphatidylethanolamine (PE). This, in turn, promoted the expression of small heat shock protein (*sHSP*) genes. The upregulation of *sHSP* genes led to reduced H_2_O_2_ production in fiber cells, inhibiting secondary wall synthesis and resulting in thinner mature fibers. These findings reveal a novel genetic strategy for manipulating fiber quality in cotton by targeting *GhROD1* and its downstream effects on lipid metabolism and *sHSP* gene expression.

## Figures and Tables

**Figure 1 ijms-25-11242-f001:**
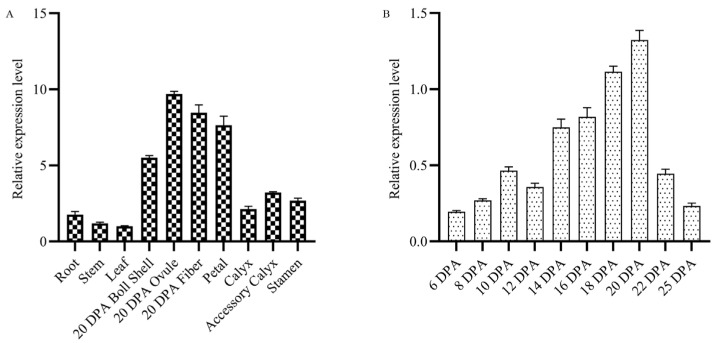
Expression of *GhROD1* in cotton plants. (**A**) Expression levels of *GhROD1* in various tissues. Total RNA was extracted from roots, stems, leaves, ovules at 20 DPA, and fibers, petals, sepals, bracts, and stamens at 20 DPA. *GhHis3* was used as an internal control. The results are presented as mean ± SD (n = 3). (**B**) Expression levels of *GhROD1* at different stages of fiber development. Total RNA was extracted from fibers at 6, 8, 10, 12, 14, 16, 18, 20, 22, and 25 DPA. The results are presented as mean ± SD (n = 3), with the *x*-axis indicating DPA.

**Figure 2 ijms-25-11242-f002:**
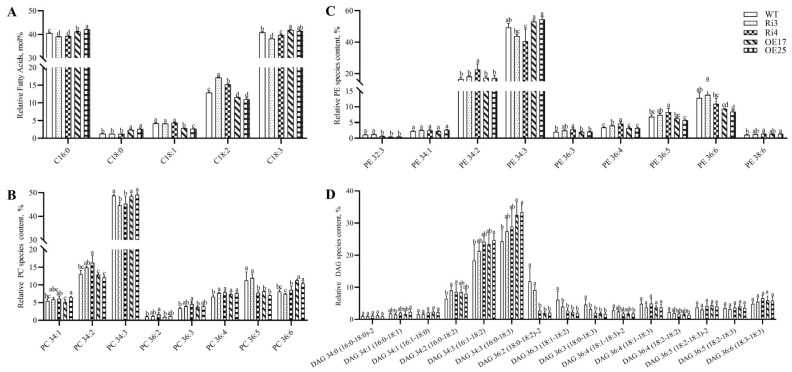
Targeted lipidomics analyses of PC, PE, and DAG in cotton fibers at the transition stage (20 DPA). (**A**) Total fatty acid composition in fibers of wild-type and transgenic cotton. C16:0: palmitic acid; C18:0: stearic acid; C18:1: oleic acid; C18:2: linoleic acid; and C18:3: linolenic acid. Data are presented as mean ± SD (n = 3). Data were analyzed by one-way ANOVA followed by Tukey’s test. Different lowercase letters indicate significant differences among groups. (**B**–**D**) Relative PC, PE, and DAG species content in the fibers at 20 DPA of wild-type and transgenic cotton. WT: wild-type (*Gossypium hirsutum* L. cv. Jimian14). Ri3 and Ri4: *GhROD1*-downregulated lines. OE17 and OE25: *GhROD1*-upregulated lines. PC: phosphatidylcholine. PE: phosphatidylethanolamine. DAG: diacylglycerol. Data for each molecular species represent the means and SDs of five independent extractions and are presented as mol% of total glycerophospholipids. Differences between transgenic lines and wild-type were analyzed by one-way ANOVA followed by Tukey’s test. Different lowercase letters indicate significant differences among groups. Components with lipid molecular species below 1% are not shown in the figure.

**Figure 3 ijms-25-11242-f003:**
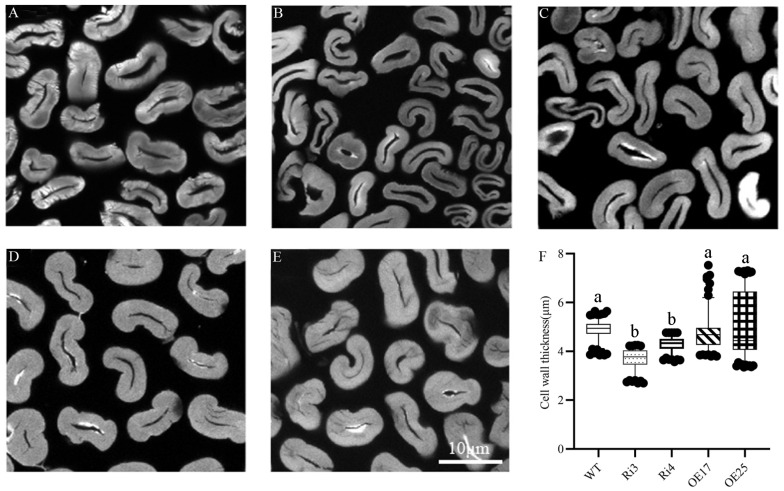
Cross-section of mature fibers in wild-type and GhROD1 transgenic cotton. (**A**–**E**) Cross-section of mature fibers in wild-type cv. Jimian14, Ri3, Ri4, OE17, and OE25. (**F**) Statistics of cell wall thickness of mature fibers in wild-type and transgenic cotton (n ≥ 150). Statistical data analysis was performed by one-way ANOVA followed by Tukey’s test. Different lowercase letters indicate significant differences among groups.

**Figure 4 ijms-25-11242-f004:**
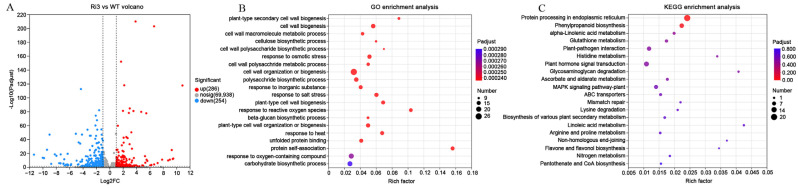
Transcriptome analysis of *GhROD1*-downregulated RNAi (Ri3) fibers at 16 DPA. (**A**) Volcano plot illustrating the differentially expressed genes (DEGs) between wild-type and Ri3 plants. In this plot, red and blue dots denote the DEGs between WT and Ri3. Red dots indicate genes that are significantly upregulated compared to WT, blue dots indicate genes that are significantly downregulated, and gray dots represent genes exhibiting no significant differential expression. (**B**,**C**) GO, KEGG enrichment analysis of DEGs between wild-type and Ri3 (bubble plot). The size of the dots represents the number of genes/transcripts associated with each GO term, showing the top 20 enrichment results. DEGs were defined by a fold change ≥ 2 and an adjusted *p*-value < 0.05.

**Figure 5 ijms-25-11242-f005:**
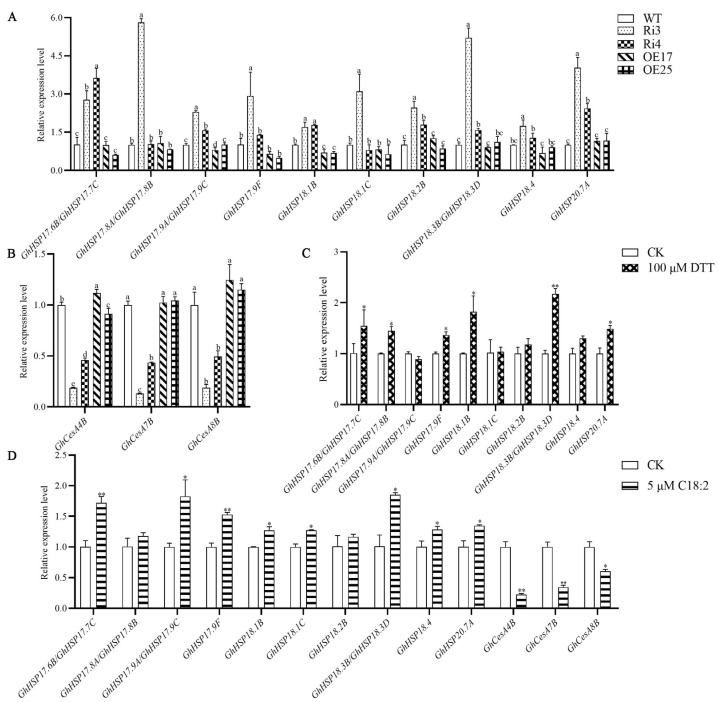
The expression profiles of *sHSPs* and *GhCesAs*. (**A**,**B**) Relative expression levels of *sHSPs* and *GhCesAs* in wild-type and transgenic cotton. Differences between transgenic lines and wild-type were analyzed by one-way ANOVA followed by Tukey’s test. Different lowercase letters indicate significant differences among groups. (**C**) Relative expression levels of *sHSPs* in ovules treated with 100 μM DTT. *GhHis3* was used as an internal control. (**D**) Relative expression levels of *sHSPs* and *GhCesAs* in fibers treated with 5 μM linoleic acid (C18:2). CK: without linoleic acid (C18:2). Results are shown as mean ± SD (n = 3). Differences between the control and treatment were analyzed using Student’s *t*-test (* *p* < 0.05; ** *p* < 0.01) for (**C**,**D**).

**Figure 6 ijms-25-11242-f006:**
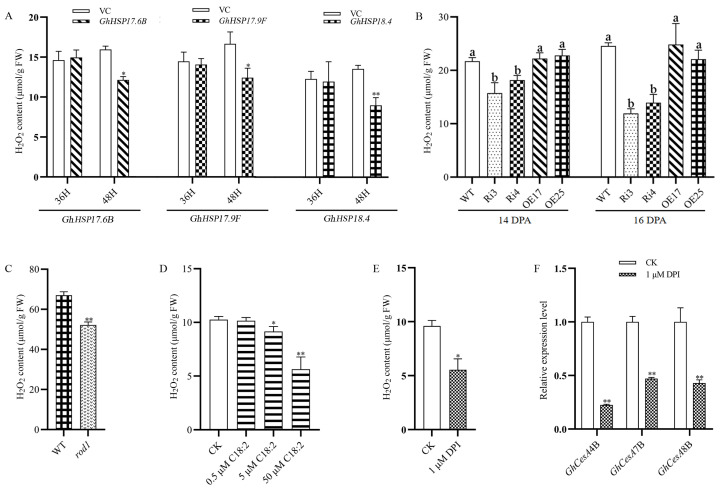
Analysis of H_2_O_2_ levels in various materials and treatments. (**A**) H_2_O_2_ content in tobacco leaves after transient expression of *GhHSP17.6B*, *GhHSP17.9F*, and *GhHSP18.4*. The empty vector (VC) served as a control. Values are the mean of three replicates, and vertical bars indicate ± SD (n = 3). (**B**) H_2_O_2_ content in wild-type and transgenic cotton fibers at 14 DPA and 16 DPA. Data were analyzed by one-way ANOVA followed by Tukey’s test. Different lowercase letters indicate significant differences among groups. (**C**) H_2_O_2_ content in siliques of wild-type *Arabidopsis* and *rod1* mutant at 9 DPA. WT: wild-type *Arabidopsis* Col-0; *rod1*: loss-of-function *rod1* mutant. Data are presented as mean ± SD (n = 3). (**D**,**E**) H_2_O_2_ content in cotton fibers treated with linoleic acid and DPI, respectively. CK: without linoleic acid or DPI as control; 0.5 μM, 5 μM, and 50 μM C18:2: supplemented with the corresponding concentration of linoleic acid in the medium. Data are presented as mean ± SD (n = 3). (**F**) Expression profiles of *GhCesA4B*, *GhCesA7B*, and *GhCesA8B* in fibers treated with DPI. CK: BT medium without DPI as control. Data are presented as mean ± SD (n = 3). Differences between the control and treatment were analyzed using Student’s *t*-test for (**A**,**C**–**F**). “*” indicates a significant difference (*p* < 0.05), and “**” indicates a highly significant difference (*p* < 0.01).

**Figure 7 ijms-25-11242-f007:**
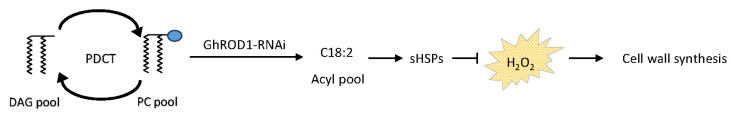
Proposed working model of *GhROD1*-mediated diacylglycerol (DAG)-to-phosphatidylcholine (PC) interconversion in cotton fiber. Downregulation of *GhROD1* can reduce the rate of DAG-to-PC interconversion, leading to the accumulation of C18:2 in acyl pool of ER, which upregulates the expression of small heat shock proteins (sHSPs). The elevated sHSPs inhibit the production of H_2_O_2_, which reduces the secondary wall synthesis, resulting in finer fibers.

**Table 1 ijms-25-11242-t001:** Comparison of fiber quality characteristics between *GhROD1* transgenic cotton and wild-type.

		Length (mm)	Strength (cN/tex)	Micronaire	Elongation Ratio (%)	Uniformity
Line No. T3 generation	WT	30.2 ± 0.4 b	28.8 ± 0.7 b	5.3 ± 0.0 a	6.8 ± 0.0 a	86.1 ± 0.7 a
	Ri3	31.3 ± 0.6 a	30.0 ± 0.7 ab	4.5 ± 0.1 c	6.8 ± 0.0 a	85.3 ± 0.7 ab
	Ri4	29.7 ± 0.6 ab	29.7 ± 0.4 ab	5.0 ± 0.0 b	6.8 ± 0.0 a	85.8 ± 0.7 a
	OE17	28.8 ± 0.6 c	29.4 ± 1.4 b	5.3 ± 0.1 a	6.8 ± 0.0 a	85.0 ± 0.9 b
	OE25	29.1 ± 0.5 c	31.1 ± 0.8 a	5.4 ± 0.1 a	6.8 ± 0.0 a	86.4 ± 0.6 a
Line No. T4 generation	WT	30.1 ± 0.4 bc	31.8 ± 1.2 b	5.3 ± 0.1 b	6.8 ± 0.0 a	84.5 ± 0.6 b
	Ri3	32.0 ± 0.5 a	35.8 ± 1.2 a	4.6 ± 0.1 d	6.8 ± 0.0 a	85.1 ± 0.4 b
	Ri4	30.0 ± 0.4 bc	32.4 ± 0.7 b	5.0 ± 0.1 c	6.8 ± 0.0 a	84.8 ± 0.7 b
	OE17	30.6 ± 0.7 b	31.5 ± 0.5 b	5.3 ± 0.1 ab	6.9 ± 0.1 a	85.4 ± 0.4 a
	OE25	29.2 ± 0.9 c	32.0 ± 0.6 b	5.4 ± 0.1 a	6.8 ± 0.0 a	85.5 ± 0.5 a
Line No. T5 generation	WT	29.3 ± 0.4 ab	31.7 ± 1.7 a	5.3 ± 0.1 b	6.9 ± 0.0 a	86.1 ± 0.8 a
	Ri3	30.8 ± 0.3 a	33.9 ± 0.6 a	4.6 ± 0.1 d	6.9 ± 0.0 a	84.6 ± 0.6 a
	Ri4	29.8± 0.4 ab	32.7 ± 1.1 a	4.8 ± 0.0 c	6.8 ± 0.0 a	84.1 ± 0.8 a
	OE17	29.4 ± 0.8 ab	31.7 ± 1.2 a	5.2 ± 0.1 b	6.9 ± 0.0 a	86.2 ± 0.3 a
	OE25	28.7 ± 0.7 b	31.7 ± 0.9 a	5.6 ± 0.1 a	6.8 ± 0.1 a	86.1 ± 0.2 a

WT: wild-type; Ri3 and Ri4: transgenic *GhROD1*-RNAi lines; and OE17 and OE25: transgenic 35S-GhROD1 lines. Three independent samples were prepared for each material. Statistical data analysis was performed by one-way ANOVA followed by Tukey’s test. Different lowercase letters indicate significant differences among groups.

## Data Availability

The data presented in this study are available on request from the corresponding author due to privacy.

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
