# Peer review of "Downregulation of the GhROD1 Gene Improves Cotton Fiber Fineness by Decreasing Acyl Pool Saturation, Stimulating Small Heat Shock Proteins (sHSPs), and Reducing H2O2 Production"

_ijms, 2024, doi:10.3390/ijms252011242_

Round 1
Reviewer 1 Report
Comments and Suggestions for Authors
The study "Downregulation of GhROD1 represses second cell wall deposition in cotton fibers through small heat shock proteins (sHSPs)-controlled H2O2 pathway" examined the downregulation of the GhROD1 gene, which led to an enhancement in fiber fineness, an increase in genes encoding small heat shock proteins (sHSPs), and a decrease in H2O2 generation. These findings have great scientific interest, but this manuscript requires major and extensive improvements and revisions.
- The title needs to be modified, focusing on the fiber quality and lipid metabolism pathways. The relationship between stress and H2O2 is universal. Thus, the current title might not attract the research community and may not explain the novelty of the research.
- The aim and objective (lines 69 to 78) in the introduction section should also be modified based on the modified title.
- The word consecutive combination of “determined showed” line 97 needs to be modified.
- The figure legend for Figure 2 needs to be modified and requires the deletion of the information related to material and methods (lines 118 to 122). Please modify and recheck all the figure legends as well.
- It is necessary to italicize the genes and scientific names throughout the manuscript, including Agrobacterium and Arabidopsis.
- The results section needs to improve extensively. It should only represent the findings and not the discussion (for example, line 128) because there is a separate discussion section. Please remove the information related to materials and methods (such as lines 129 to 131) and discussion that are mentioned in the result section.
- The footnote in Table 1 (lines 161 to 163) needs to be modified as it contains information related to materials and methods.
- The inclusion of a separate Conclusion section might be better for this article, and the reader will get the overall concluding remarks with some future directions.
Overall, the manuscript required major and substantial corrections, as well as some spell-checking.
Comments on the Quality of English LanguageModerate editing required.
Reviewer 2 Report
Comments and Suggestions for Authors
The manuscript presents study aimed to show the involvement of lipid metabolism in cotton fibres formation. The problem has not been searched intensively before and seems to be relevant both from theoretical and practical fibre production aspects. The authors attempted to reveal mechanisms at molecular and biochemical levels which might be implicated in shaping the quality of cotton fibres, the main natural fibre source. The plant material was obtained in multi-seasonal field experiments. The authors used diverse commonly accepted advanced modern techniques . Possible sequence of events is proposed: down regulation of gene coding phosphatidylcholine diacylglycerol cholinephosphotransferase increases contribution of linoleic acid in fibres that have the impact on expression of genes coding heat shock proteins and consequently the hydrogen peroxide production. The latter leads finally, through a signalling pathway, to altered cellulose biosynthesis and fibre quality.
|
Manuscript presents extensive methodically study which includes creation of transgenic lines, searching transcriptomes, biochemical analysis of fatty acid content and profile, anatomical measurement and fibre quality evaluation using routine technological approach. The expression of GhROD1 gene, key in the study, was monitored across various organs/tissue and developmental stages. The results derived from molecular studies were also supported by ovule culture experiments with supplementation of fatty acid component to the medium. Overall, the manuscript presents high quality studies in majority of aspects and well fits with the scope of Int. J. Mol. Sci.. |
The manuscript is well designed and written showing excellent competence of the Authors.
However, statistical analysis applied in this study raises my considerable reservations. If the statistical evidence comes from incorrect analysis/models the results (any comparisons) are uncertain. This might make presented here scientific inferences questionable.
The authors used mainly Student’s t -test (with a few exceptions) to compare means. However, it has limited applicability and makes it suitable only for comparison of means of two groups in data which presents normal distribution. When more groups are compared at the same time - different statistical approach is needed ( e.g. ANOVA and post-hoc tests or non-parametric tests).
Some detailed comments:
Figure 1
Comment: No statistical comparisons of means in Fig 1?
Figure 2.
Comment: We have here rather multiple comparisons vs WT, so Student test cannot be used. Moreover, these data are percentage data. It need more advanced statistics than ANOVA.
Table 1.
Comment: Multiple comparisons were performed to compare means vs WT. Thus, the Student’s test is not appropriate.
Figure 3D
Comment: Similar issue as above
Figure 4
Comment: The Y-axis labels are rather illegible.
Figure 5
Comment: We have multiple comparisons va WT (A-B), the issue as mentioned above. Moreover, the data are expressed as relative values, so neither Student’s test nor ANOVA should be applied (A-D).
Please refer to discussion in https://www.researchgate.net/post/Ttest-for-qPCR-data-on-deltaCt-or-on/5f06e00e2f1885300f1fd209/citation/download
or other similar or appropriate papers dealing with similar data.
Table 1. Elongation ratio (%)
Comment: decimals should be the same for the mean and SD.
Line 383
" …H2O2, which deduces the secondary wall synthesis….."
Comment: probably typographic error?
Reviewer 3 Report
Comments and Suggestions for Authors
Manuscript ID: ijms-3229386 - Review Report
Please find my review for the manuscript ID: ijms-3229386 entitled as “Ding et al., Downregulation of GhROD1 represses second cell wall deposition in cotton fibers through small heat shock proteins (sHSPs)- 3 controlled H2O2 pathway”
The manuscript is well written. The introduction part clearly reviewed with coherent information. Pertinent data with figures were generated. The results and discussions were also well-articulated with sound interpretation.
I would like to mention the following issues need to be amended.
1. The legends of almost all figures of the manuscript are too long with too much explanation and information those should be incorporated in materials and methods. Furthermore, those lengthy figures legends were a repetition of materials and methods. So, it is better to re-write in short precise way. For example: legends of Figure 2. Line#107-126 and Figure 6 legends from line#259 to 285. The same holds true for supplementary figures S2, S4 and Table 1 description parts (line#159-165).
2. There are no significant differences between overexpression transgenic lines and wild type for the parameters mentioned in Table 1 as compared to downregulated transgenic lines. What is the scientific justification?
3. Why the authors used different destination vectors pCB2012 and pCB2016 for GhROD1 and GhHSP ectopic expression, respectively? See lines #386 and 403.
Thank you!
Round 2
Reviewer 1 Report
Comments and Suggestions for Authors
Thank you so much for modifying and improving the manuscript. The title needs to be further modified. It could be "Downregulation of the GhROD1 gene Improves Cotton Fiber Fineness by Decreasing Acyl Pool Saturation, Stimulating Small Heat Shock Proteins (sHSPs), and Reducing H2O2 Production" or "Downregulation of GhROD1 improves cotton fiber fineness by decreasing acyl pool saturation, stimulating Small Heat Shock Proteins (sHSPs), and lowering H2O2 production".
